# Chloroquine Enhances Death in Lung Adenocarcinoma A549 Cells Exposed to Cold Atmospheric Plasma Jet

**DOI:** 10.3390/cells12020290

**Published:** 2023-01-12

**Authors:** Ekaterina Patrakova, Mikhail Biryukov, Olga Troitskaya, Pavel Gugin, Elena Milakhina, Dmitriy Semenov, Julia Poletaeva, Elena Ryabchikova, Diana Novak, Nadezhda Kryachkova, Alina Polyakova, Maria Zhilnikova, Dmitriy Zakrevsky, Irina Schweigert, Olga Koval

**Affiliations:** 1Institute of Chemical Biology and Fundamental Medicine, Siberian Branch of the Russian Academy of Sciences, 630090 Novosibirsk, Russia; 2Department of Natural Sciences, Novosibirsk State University, 630090 Novosibirsk, Russia; 3Khristianovich Institute of Theoretical and Applied Mechanics, Siberian Branch of the Russian Academy of Sciences, 630090 Novosibirsk, Russia; 4Rzhanov Institute of Semiconductor Physic, Siberian Branch of the Russian Academy of Sciences, 630090 Novosibirsk, Russia; 5Department of Radio Engineering and Electronics, Novosibirsk State Technical University, 630073 Novosibirsk, Russia

**Keywords:** anticancer approaches, cold atmospheric plasma jet, cell death modalities, autophagy, lysosomes, chloroquine, cathepsin D, mitochondrial quality

## Abstract

Cold atmospheric plasma (CAP) is an intensively-studied approach for the treatment of malignant neoplasms. Various active oxygen and nitrogen compounds are believed to be the main cytotoxic effectors on biotargets; however, the comprehensive mechanism of CAP interaction with living cells and tissues remains elusive. In this study, we experimentally determined the optimal discharge regime (or semi-selective regime) for the direct CAP jet treatment of cancer cells, under which lung adenocarcinoma A549, A427 and NCI-H23 cells demonstrated substantial suppression of viability, coupled with a weak viability decrease of healthy lung fibroblasts Wi-38 and MRC-5. The death of CAP-exposed cancer and healthy cells under semi-selective conditions was caspase-dependent. We showed that there was an accumulation of lysosomes in the treated cells. The increased activity of lysosomal protease Cathepsin D, the transcriptional upregulation of autophagy-related MAPLC3B gene in cancer cells and the changes in autophagy-related proteins may have indicated the activation of autophagy. The addition of the autophagy inhibitor chloroquine (CQ) after the CAP jet treatment increased the death of A549 cancer cells in a synergistic manner and showed a low effect on the viability of CAP-treated Wi-38 cells. Downregulation of Drp1 mitochondrial protein and upregulation of PINK1 protein in CAP + CQ treated cells indicated that CQ increased the CAP-dependent destabilization of mitochondria. We concluded that CAP weakly activated pro-survival autophagy in irradiated cells, and CQ promoted CAP-dependent cell death due to the destabilization of autophagosomes formation and mitochondria homeostasis. To summarize, the combination of CAP treatment with CQ could be useful for the development of cold plasma-based antitumor approaches for clinical application.

## 1. Introduction

The cold atmospheric plasma jet is a promising tool for anticancer therapy. It is currently under intensive development in order to translate it to clinical use [1,2]. A plasma jet appears visually continuous, but actually consists of a sequence of millimeter-sized streamers (ionization spots) propagating from the plasma device to the biotarget. To date, many studies have shown that low temperature atmospheric plasma affected the viability of cancer cells via various reactive species of oxygen and nitrogen (ROS and RNS) that were formed in ionized inert gas flow [3,4,5]. Differences between the effects of plasma on healthy cells and tumor cells also remain a matter of debate. The efficiency of antitumor CAP treatment, which is associated with CAP toxicity to cancer cells, depends on the chosen parameters of the plasma device operation, such as voltage frequencies and amplitudes, the composition of the working gas and its pumping rate, the dimensions of the nozzle-target gap and the type of target [6,7,8,9,10]. The optimization of irradiation modes is aimed at increasing the cytotoxic effect on tumor cells without affecting healthy cells. Under optimal irradiation conditions, it is possible to analyze which factors, in particular, provide this selectivity at the molecular level. Moreover, the type of cell death occurring under optimized irradiation parameters, which stimulate only tumor cell death, may differ from the death type induced by CAP, with parameters aimed at killing cells in general. This is possible because optimized irradiation conditions, i.e. those that are semi-selective for cancer cells, should be more gentle, and the stress response of tumor cells may also be reduced and less effective in activating certain death pathways. This is possible because optimized irradiation conditions, i.e. those that are semi-selective for cancer cells, should be more gentle, and the stress response of tumor cells may also be reduced and less effective in activating certain death pathways [11]. The difference in sensitivity to CAP-dependent reactive oxygen species may be attributed to different basal levels of oxidative stress, as well as the activity of antioxidant systems, in healthy and cancer cells [12,13]. Various cell death modalities, such as apoptosis, necroptosis, autophagy and ferroptosis, have been described for cells irradiated by CAP or plasma-activated medium (PAM) [14,15]. The overcoming of oxidative stress, which leads to the accumulation of damaged organelles, can occur via the autophagy pathway, aimed at cell survival. Indeed, it was recently demonstrated that excessive ROS and oxidative stress could activate autophagy [16]. Autophagy is a metabolic process which enables cells to adapt to stressful conditions; it enables cancer cell survival under hypoxia and nutrient [17]. However, the pro-death role of autophagy is complicated due to the extensive cross-talk between different signaling pathways [18]. In tumors, autophagy can promote or suppress their progression in a context-dependent manner, when survival or death mechanisms are activated. Thus, the regulation of autophagy has promising applications in the development of new antitumor approaches. Therefore, it was important to determine whether autophagy activation occurred in tumorous and normal cells after CAP exposure in our model systems. Opposing data have been published regarding whether cold plasma irradiation does or does not activate autophagy. Conway et al. demonstrated that CAP treatment of U373 glioblastoma cells induced an accumulation of lysosomes with no autophagy occurring, whereas Yoshikawa et al. showed that a plasma-activated medium promoted autophagic death of endometrial cancer cell lines [19,20]. Preventing pro-survival autophagy by chemically inhibiting this process often enhances the cytotoxic and selective effects of antitumor therapy.

One of the most widely used autophagy inhibitors is chloroquine. Chloroquine (or hydroxychloroquine) is a well-characterized drug with a low toxicity, which is available for autophagy inhibition in clinical studies [21]. Chloroquine (CQ) is accumulated in autophagosomes, preventing the fusion of autophagosome and lysosome, which results in the lack of a productive degradation of nutrients [22]. Moreover, CQ was demonstrated to sensitize cancer cells to conventional or potential chemotherapeutics [23,24].

In this study, an optimal CAP jet mode was found and applied in our biological experiments in order to analyze cytotoxic selectivity, with respect to cancer and normal cells of the same histological origin. Chloroquine was used, together with a CAP jet, in order to modulate the killing effects of the treatment. The combination reinforced the death rate of cancer cells treated with CAP. In addition, we reported that it was unlikely that cancer cells exposed to CAP died through the pathway of autophagy.

## 2. Materials and Methods

### 2.1. Cell Cultures

A549 human lung carcinoma cells (purchased: ATCC # CCL-185), Wi-38 (purchased: ATCC # CCL-75) and MRC-5 lung fibroblasts (ATCC # CCL-171, from Flow Laboratories, Oldham, UK) were grown in DMEM: Nutrient Mixture F-12 (DMEM:F12, Sigma-Aldrich, St. Louis, MO, USA) supplemented with 10% fetal bovine serum (GIBCO, Thermo Fisher Scientific, Waltham, MA, USA), 2 mM L glutamine, 250 mg/mL amphotericin B and 100 U/mL penicillin/streptomycin in 5% CO_2_. NCI-H23 epithelial-like lung adenocarcinoma (ATCC # CRL-5800) and A427 lung carcinoma cells (ACC 234, Leibniz Institute DSMZ, Brunswick, Germany) were grown in RPMI 1640 (GIBCO, Thermo Fisher Scientific, Waltham, MA, USA) supplemented with 10% fetal bovine serum (Sigma-Aldrich, St. Louis, MO, USA), 2 mM L glutamine, 250 mg/mL amphotericin B and 100 U/mL penicillin/streptomycin. Cells were maintained as previously described [25].

### 2.2. Experimental Plasma Jet Setup and Plasma Jet Simulation Details

In our experiments, the source of the plasma jet was a coaxial dielectric channel with an inner diameter of 8 mm and a capillary nozzle with a diameter of 2.3 mm [8,9,26]. A schematic diagram of the device is shown in Figure 1a. The discharge zone in the dielectric tube was formed by the powered electrode inside and the grounded ring outside of the tube. Culturing plates were placed on the table at 25 mm apart from the nozzle. The plasma jet was generated in the laminar flow of helium which was pumped through the dielectric channel of the plasma device. The typical conditions for CAP irradiation of the cells were as follows: the voltage amplitude, U = 2.85–3.75 kV, the voltage frequency, f_U_ ≈ 50 kHz, the helium pumping rate, υ = 9 L/min and the nozzle-target gap, d = 1–3.5 cm.

### 2.3. Cell Viability Assay

Cell viability was detected 24 h after CAP irradiation or drug treatment using an MTT test and a real-time iCELLigence system (ASEA Biosciences, San Diego, CA, USA), as described previously [25]. The effects of the combination treatment were calculated with CompuSyn software version 1.0 (ComboSyn, Inc., Paramus, NJ, USA). The MTT assay data of the mono-treatment and combination treatment were used for the calculation of a combinatorial index (CI) by the software. Synergism (CI < 1) was quantified using the Chou–Talalay method [27].

### 2.4. Cell Death Analysis with Annexin V/PI-Staining

Treated and control cells were cultured under standard cultivation conditions prior to analysis. Next, spontaneously detached cells were collected and combined with cells detached using Trypsin-EDTA (Gibco, Thermo Fisher, Waltham, MA, USA). A Trypsin inhibitor from soybean was used to inhibit trypsin-initiated proteolysis. Cell samples were incubated with components of the FITC Annexin V Apoptosis detection kit (BD Biosciences, San Jose, CA, USA) as recommended by the manufacturer. Cell death was estimated by flow cytometry using FACScantoII (BD Biosciences, Franklin Lakes, NJ, USA). Cells were initially gated based on forward versus side scatter to exclude small debris, and ten thousand events from this population were collected. Data analysis was performed by FACSDiva Software Version 6.1.3 (BD Biosciences).

### 2.5. Measurement of ROS

Following treatment with CAP, cells were stained with DCFDA (10 μM). Fluorescently-labeled cells were analyzed by flow cytometry for quantification signal in the FITC channel (λ_ex_ = 491 nm, λ_em_ = 525 nm). The cells used for analysis were stained immediately after the CAP treatment (for 30 min time point) or 30 min before the analysis (for other time points).

### 2.6. RNA-Seq and Transcriptome Analysis

Cells were lysed by Trizol™ (Life Technologies, Carlsbad, CA, USA) according to the manufacturer’s protocol. Each sample was duplicated. The DNA libraries were constructed according to standard Illumina recommendations. The sequencing was done on the Illumina 1500 NextSeq platform. Sequencing was performed at the Interdisciplinary Center of Collective Use of the Institute of Fundamental Medicine and Biology of Kazan (Volga Region) Federal University (Kazan, Russia). Bioinformatic analysis of high-throughput RNA sequencing data was performed using the following software: Trimmomatic version v0.32 [28] to remove adapter sequences; bowtie2 to filter rRNA, tRNA and snRNA, SINE-, LINE- and DNA-repeat consensus sequences; (RepBase [29] accessed on 14 January 2022; Genetic Information Research Institute, Mountain View, CA, USA) for low-complexity simple repeats, as well as mitochondrial DNA (NC_012920.1); STAR 2.7.1a (ENCODE, Stanford University, Stanford, CA, USA) [30] to align experimental sequences with human genome/transcriptome assembly hg19 sequences (NCBI RefSeq); QoRTs v1.3.6 [31] to analyze alignment quality and quantify individual RNA contributions; DESeq2 1.36.0 [32] (R version 4.1.3 and Bioconductor 3.14) for differential analysis of gene expression.

### 2.7. Western Blot

Cells were lysed with cell lysis buffer (50 mM Tris, pH 8.0, 5 mM EDTA, 150 mM NaCl) containing 0.1% SDS, 1x complete protease inhibitor cocktail (Roche Diagnostics GmbH, Mannheim, Germany) and 1 mM PMSF. Western blot analysis of cell lysates was performed as described previously [24] using the following antibodies: anti-LC3B and anti-ATG5 (Abcam, Cambridge, UK), anti-p62/SQSTM (Sigma-Aldrich, St. Louis, MO, USA), anti-BECN1 (SAB4100184-200UL, Sigma-Aldrich), anti-PINK1 [N4/15] (Abcam, Cambridge, UK), anti-DRP1 [3B5] (Abcam, Cambridge, UK), anti-COX IV (Abcam, Cambridge, UK), anti-b-Tubulin (Sigma-Aldrich, St. Louis, MO, USA), anti-polyclonal rabbit-anti-mouse and mouse-anti rabbit HRP-conjugated secondary antibodies (Biosan, Novosibirsk, Russia).

### 2.8. Cellular Fractionation

Cellular fractionation was performed as described previously [33]. In total, 1 × 10^6^ cells were trypsinized and washed in PBS. All centrifugation steps in the following fractionation were performed at 17,000× *g*. A swelling step was performed in swelling buffer (10 mM HEPES Ph 7.6, 10 mM KCl, 2 mM MgCl2, 0.1 mM EDTA, Protease Inhibitor mix) for 5 min, followed by addition of 0.3% NP-40 (Thermo Fisher Scientific Inc., Waltham, MA, USA) for 1 min. Centrifugation for 1 min resulted in a separation of the cytoplasmic fraction. The remaining pellet was resuspended in 100 µL of swelling buffer and centrifuged for 15 s. The pellet was incubated for 30 min with 40 µL of a nucleus buffer (50 mM HEPES pH 7.8, 50 mM KCl, 300 mM NaCl, 0.1 mM EDTA, 10% Glycerol, Protease Inhibitor mix), resuspended every 10 min and centrifuged for 5 min. The nuclei-containing supernatant and mitochondria-containing pellet were separated, and the pellet was washed twice with PBS and dissolved in 1× Laemmly Sample Buffer (Biorad, Hercules, CA, USA).

### 2.9. Cathepsin D Activity Assay

Cells (2 × 10^3^) were grown in 96-well plates under standard conditions for 24 h. After that, cells were exposed to CAP for 1 min (AC voltage frequency of 50/4 kHz and the amplitude of 3.5 kV). Cells continued to be cultivated for 2 h or 5 h, then cells were lysed as recommended by the manufacturer of the Cathepsin D activity assay kit (Abcam, Cambridge, UK, # ab65302). Cathepsin D activity in the cell lysates was assayed by a fluorescent-based method that utilized the preferred Cathepsin-D substrate sequence GKPILFFRLK(Dnp)-D-R-NH2) labeled with methyl cumaryl amide. The results were analyzed using a fluorometer (Cary Eclipse, Varian, Australia) at Ex/Em = 328/400 nm with 5 nm excitation and emission slits.

### 2.10. Lysosomes Content Analysis with Acridine Orange

CAP-irradiated or control cells, growing in 96-wells plates under standard conditions, were detached from the plastic substrate with StemPro™ Accutase™ (GIBCO, Thermo Fisher, Waltham, MA, USA), washed twice by PBS and stained with acridine orange (1 µg/mL) (Thermo Fisher Scientific, Waltham, MA, USA) for 15 min at room temperature in the dark. Next, cells were washed by PBS and analyzed by flow cytometry in the APC fluorescence channel. Initially, cells were gated based on forward versus side scatter to exclude small debris, and ten thousand events from this population were collected.

### 2.11. Lysosomes Content Analysis with LysoTracker

LysoTracker™ Red DND-99 (Thermo Fisher Scientific, Waltham, MA, USA) (ex/em 577/590 nm) was added to the CAP-irradiated or growing control cells 1.5 h before the analysis. Subsequently, cells were trypsinized and washed twice in PBS. Cells were analyzed by flow cytometry. Cells were initially gated based on forward versus side scatter to exclude small debris, and ten thousand events from this population were collected. LysoTracker signals were detected in APC fluorescence channel.

### 2.12. Transmission Electron Microscopy

Treated and control cells were detached by 0.5% Trypsin-EDTA, diluted by L-15 medium (1 mL) with 10% FBS and centrifuged for 5 min at 3000 rpm. Cells were collected by centrifugation and cell pellets resuspended in 4% paraformaldehyde, then additionally fixed with 1% osmium tetroxide solution. Sequential dehydration block sections were performed, as described previously [24]. Contrasted ultrathin sections were examined in a transmission electron microscope JEM 1400, JEOL (Tokyo, Japan) equipped with a digital camera, Veleta, EM SIS (Muenster, Germany).

### 2.13. Statistical Analysis

All experiments were performed at least three independent times. Data shown are presented as mean ± SD unless stated otherwise. A Student’s *t*-test was used to compare effects between two groups. A *p*-value of less than 0.05 was considered significant.

## 3. Results

### 3.1. Finding CAP Exposure Regimens for Predominant Death Induction in Cancer Cells

To determine the conditions of CAP irradiation under which helium CAP exposure predominantly killed cancer cells, five lung-originated cell lines were used: lung adenocarcinoma A549 cells, NCI-H23 epithelial-like lung adenocarcinoma, A427 carcinoma cells and normal lung fibroblasts, as well as Wi-38 and MRC-5 lung fibroblasts. In the experiments, streamers appeared near the powered electrode and began to propagate at each positive half-cycle of the alternating voltage (Figure 1a). In this study, we used the CAP jet generated with voltage frequency of 50 kHz and amplitude of 3.5 kV. In this case, only every fourth streamer approached the target. Thus, the frequency of the target being touched by a streamer was f_I_ = 50/4 kHz. A study of target heating occurred under the following conditions: 50 kHz CAP AC voltage, U = 2.85–3.8 kV and helium flow υ = 9 L/min. It showed that the target remained cold (T < 40 °C) in the range U = 2.85–3.5 kV (Figure 1b). Taking into account the biosafe temperature regime and the results of irradiation on A431 cancer cells and arbitrary normal HEK293 cells in our previous work, the initial treatment conditions were as follows: CAP generation AC voltage frequency of 50 kHz (f_I_ = 50/4 kHz) and amplitude of 3.5 kV [8].

By varying the duration of CAP irradiation from five seconds to two minutes, we showed a time-dependent decrease of cell viability for cancer cells as well as normal cells (Figure 2a,b). A549 cells were more sensitive to the killing activity of CAP. CAP treatment for 2 min completely killed A549 cells, led to the appearance a lot of debris and small, round, detached floating cells. This treatment also decreased the number of cells with healthy morphology (Figure 2c). Real-time cell analysis of A549 cells verified the extremely high cytotoxic activity of CAP exposure for 2 min. Irradiation for 60 s resulted in a difference in viability of tumor cells and healthy cells that was up to 40%—which allowed us to describe these conditions as optimized semi-selective irradiation conditions for tumor cell death.

To summarize, the specified irradiation parameters and a duration of 60 s were considered to be semi-selective for tumor cell death and were used in further experiments to evaluate the molecular changes in CAP-exposed cells.

### 3.2. The CAP-Dependent Increase of Intracellular ROS

Variation of the CAP exposure parameters affected the amount of reactive oxygen species generated by the plasma jet that triggered oxidative stress [34,35,36]. We compared the dynamic of the amount of intracellular ROS after semi-selective CAP exposure of A549 and Wi-38 cells (1 min) by flow cytometry using the ROS-sensitive fluorescent indicator DCFDA. 2′,7′-Dichlorodihydrofluorescein diacetate (H2DCFDA, DCFDA) was converted inside the cell to 2′,7′-dichlorofluorescein (DCF-, λ_ex_ = 485 nm and λ_em_ = 530 nm) by intracellular reactive oxygen species. CAP exposure resulted in an increase of DCF- signal, indicating an ROS increase in both cancer and normal cells (Figure 3). However, we didn’t find a difference in the early ROS response between cancer and normal cells under semi-selective conditions. The tendency of ROS to increase in A549 cancer cells led to a stronger difference in normal and cancer cells four hours after CAP exposure. Thus, an elevated ROS accumulation in cancer cells after CAP irradiation positively correlated with cell death rate, as evaluated.

### 3.3. Pan-Caspase Inhibitor Z-Vad Markedly Suppresses CAP-Dependent Cell Death

To analyze if the death of CAP-exposed A549 and Wi-38 cells under semi-selective conditions was caspase-dependent, we used pan-caspase inhibitor Z-VAD-FMK (z-Vad). Cells were CAP-irradiated for 1 min and 24 h later they were stained by AnnexinV/PI to estimate cells in early and late apoptosis/necrosis. Z-Vad was added to the cells two hours before the CAP treatment. Z-Vad concentration was chosen according to its cytotoxicity, under which Z-Vad improved viability of A549 cells (Figure 4a and Appendix A). Flow cytometry analysis showed that Z-Vad partly suppressed cell death activity of CAP (Figure 4). Thus, caspase-dependent cell death made a major contribution to CAP-dependent cell killing.

Nevertheless, the question remained whether autophagy developed after CAP irradiation and whether it contributed to cell death. Usually, autophagy aims at overcoming stress conditions for cell survival; however, in some cases, autophagy accelerates cell death [37]. Therefore, we focused this study on autophagy signatures in cells after CAP irradiation.

### 3.4. The Influence of Autophagy Inhibitor Chloroquine (CQ) on the Cytotoxic Effects of CAP

Recently, it has been demonstrated that ROS can induce autophagy, leading to the death of cancer cells, meaning that autophagy may contribute to CAP-induced cell death [20]. Therefore, we hypothesized that CAP-dependent ROS accumulation may also stimulate autophagic cell death in treated cells. There is growing evidence that, in specific contexts, autophagy can indeed facilitate cell death. Cell death by autophagy can be defined as cell death that is independent of apoptosis, which can be blocked by pharmacological or genetic inhibition of autophagy and which results in an increase in autophagic flux (completion of autophagy), rather than solely increased detection of markers of autophagy [18]. In the case of therapeutic drugs or treatments, pro-survival autophagy can be considered a negative factor and pro-death autophagy a positive factor. When developing an antitumor approach, the use of the most appropriate partner drug can greatly enhance the therapeutic effect. Such a partner could be a drug that inhibits pro-survival autophagy.

Lung adenocarcinoma A549 cells and normal lung fibroblasts Wi-38 were exposed to CAP under the semi-selective conditions, as determined above. The autophagy inhibitor CQ was added to the cells immediately after irradiation. The effects of CQ on control and CAP-irradiated cells were analyzed by a MTT viability test (Appendix A Appendix A). MTT analysis showed that CQ more significantly enhanced the cell death rate in A549 irradiated cancer cells (Figure 5a). The cytotoxic effect of the combination of CQ and CAP jet exposure was calculated as described in Methods. The combinatorial index (CI) for combination of these two approaches was estimated as CI = 0.60, and this value defined the synergism (CI < 1) for this treatment regime. Synergism implies an amplified effect, compared to the simple sum of two effectors.

The finding that inhibition of autophagy enhanced CAP-initiated cell death could indirectly indicate the involvement of autophagy in CAP-dependent cytotoxicity.

### 3.5. Lysosomal Response to CAP Treatment and to Autophagy Inhibition

To analyze whether CAP exposure stimulated autophagy-related changes of lysosomes in irradiated cells, we used acridine orange (AO) staining. Lysosomes are key cellular organelles involved in autophagy for maintaining cell health [38]. Acridine orange is a fluorescent dye which is accumulated in acidic lysosomes. Its red fluorescent signal reflects the number of functionally-active lysosomes. CAP treatment resulted in the shift of the AO signal in the samples of A549 cells (Figure 5b, P4 region). We observed that CQ stimulated the increase of the intensity of red signal in both cancer A549 and Wi-38 normal cell lines (Figure 5c); however, in cancer cells, the increase was more substantial. These data reflected a stronger cellular accumulation of lysosomes, which were restricted from fusing with autophagosomes, in cancer cells compared to control cells. After the CAP jet exposure, the autophagy-related red signal of AO increased in A549 cells, which could reflect the increase of lysosomes. Semi-selective (1 min) CAP irradiation enhanced the lysosome-related signal up to three times in cancer cells, compared with non-treated cells, while in healthy Wi-38 cells, this signal increased weakly.

To further confirm this hypothesis, we used a lysotracker. LysoTracker Red DND-99 is a cell-permeable red fluorescent dye that stains acidic compartments within a cell, such as lysosomes. We found that the lysotracker signal increased in CAP-treated cell samples. Only CAP-treated tumor cells showed a significant increase in signal relative to control (Figure 5d). Since CQ changed the pH of the lysosomes from acidic to neutral (≈7.4), the data for the samples with CQ were, as expected, lower than the control values. Thus, it was demonstrated that the exposure of cells to CAP led to an increase in lysosomes. The observed decrease in lysosome signal in cells treated with CQ, or the combination of CAP + CQ relative to control tumor cell samples, allowed us to make an indirect conclusion about a decrease in the level of productive autophagy in these cells. Thus, the autophagy suppression by CQ may have resulted in the decrease of cell viability by reducing the basic level of autophagy necessary for normal cell function.

We examined numerous ultrathin sections of CAP-treated and control A549 cells by TEM (Figure 6) and analyzed a representative amount of lysosome structures in both cases. We paid special attention to the analysis of structures associated with the formation of autophagosomes. However, no significant increase in amount or changes in morphology of autophagosomes and lysosomes in CAP-treated cells (as compared to control cells) were detected.

Thus, the combination of CAP irradiation and CQ significantly promoted cell death, despite the fact that CAP weakly stimulates the increase of lysosomes.

### 3.6. Cathepsin D Activity after CAP Exposure

In lysosomes and autophagolysosomes, the acidic proteases digest the internalized waste cell proteins and peptides, maintaining cell health [39]. One such essential protease is the aspartyl protease cathepsin D, which is involved in cytoprotection against excessive aggregated proteins in addition to being engaged in various cell death cascades [40]. We further defined the activation of cathepsin D following exposure to CAP on A549 and Wi-38 cells using a fluorometric technique. We found that CAP exposure only activated CatD in cancer A549 cells (Figure 7). We suggested that such increase in CatD activity was associated with an increase in the number of lysosomes in tumor cells after CAP irradiation, as demonstrated above. It is possible that the differences in CAP-stimulated CatD activity could have contributed, in specificity, to CAP irradiation toward cancer cells.

### 3.7. The Differences in Autophagy-Related Gene Responses to CAP Exposure in Cancer and Healthy Cells

We analyzed changes in the mRNA expression of genes whose proteins are involved in the activation and execution of autophagy, namely: the ATG-family, LC3B, p62 and Beclin1. A459 and Wi-38 cells were irradiated by CAP under semi-selective conditions (1 min). Then, after 3 h and 24 h, the cells were lysed and samples were prepared for transcriptome analysis, as described in Section 2 (Materials and Methods). The differential gene expression data were analyzed using software packages; from those, the curves of individual mRNA levels in the samples were plotted (Figure 8).

Two time points allowed us to estimate early (3 h) and late (24 h) post-treatment responses. We observed that the types of transcription changes of the studied genes (upregulation or downregulation), in general, were similar for both cell lines, with some exceptions. MAP1LC3B (microtubule-associated protein light chain 3B protein, LC3B) gene transcription was upregulated in A549 cells, with no significant changes in Wi-38 cells 24 h after the CAP irradiation. A two-fold increase detected in LC3B mRNA levels 24 h after irradiation could have indirectly reflected a tendency toward autophagy activation in A549 cells. However, no significant changes in the other autophagy-related genes were found to unequivocally indicate the activation of autophagy. Thus, we were able to conclude that CAP caused minor changes in the expression of autophagy-related genes in healthy and cancer cells.

### 3.8. Autophagy-Related Protein Responses to CAP Treatment

In general, autophagy is regulated at the level of protein activity. Thus, changes in the major ensemble of proteins are important for the analysis of autophagy activation in cells. For example, one of the signs of autophagy is the appearance of a processed form of LC3B protein, LC3IIB, without increases in the total amount of LC3B. To analyze changes of LC3B, p62 and ATG5—which are players on the autophagic team, so to speak—A549 cells and Wi-38 cells were irradiated by CAP. Then, 6 h and 24 h later, cell lysates were analyzed by Western blot (Figure 9). We also monitored these proteins in the cells, subjected ro combinatorial treatment of CAP and CQ. The protein composition of CAP mono-treatment samples revealed slight increases in autophagy, relative to the baseline (control) level. However, combinatorial treatment of CAP and CQ resulted in increased LC3B-II form and decreased p62—the signs of autophagy. Comparing the mono-irradiation mode and the combination of CQ and CAP treatment, the combinatorial mode led to a concerted activation of protein markers of autophagy. Moreover, we revealed differences in protein responses between healthy Wi-38 cells and A549 lung adenocarcinoma cells for the combinatorial treatment. We observed up to a tenfold increase of LC3B-II in A549 cells in comparison with control cells. Meanwhile, in Wi-38 cells, the LC3B-II form increased, but less drastically. Functionally, CQ prevented fusion of lysosomes with autophagosomes, abolishing the degradation of the autophagosome’s cargo. Thus, autophagy was unproductive in CQ + CAP-treated cells. This unproductive autophagy, with lysosome accumulation, shifted pro-survival processes toward a cell death outcome.

### 3.9. CAP and CQ Change the Proteins Associated with Mitochondrial Quality Control

Cytotoxicity of CQ in CAP-treated A549 cells could essentially be due to a decrease in mitochondrial quality and bioenergetic function in A459 cells. Indeed, it was shown that inhibition of autophagy with chloroquine decreased mitochondrial quality and bioenergetic function in primary neurons [41]. We analyzed PINK1 and Drp1 proteins in the mitochondrial fraction (Appendix A Appendix A) of treated cells by Western blot. PINK1 protects cells from damage-mediated mitochondrial dysfunction, oxidative stress and cell apoptosis/PINK1 by recruiting the Parkin to mitochondria to initiate mitophagy. The PINK1 level is definitely regulated in the healthy steady state of mitochondria [42]. Drp1 is required for mitochondrial fission and acts as a protein marker for mitochondrial dynamics [43]. We found that CAP irradiation, as well as CQ treatment or CAP + CQ treatment, of A549 cells led to a decrease of Drp1 levels (Figure 10 and Appendix A). The decrease of Drp1 resulted in impaired mitochondrial fission. Moreover, a significant increase in PINK1 in a sample of A549 cells treated with CAP and CQ was observed 24 h after the treatment. Thus, according to the indicated features, CQ enhanced the destabilization of mitochondrial quality control in CAP-treated A549 cells.

## 4. Discussion

In this study, we tried to experimentally determine the cell irradiation parameters leading to the death of tumor cells only. We showed that, under CAP parameters with a voltage amplitude of 3.5 kV and current frequency 50/4 kHz, working gas—helium 9 L/min, and treatment duration 60–90 s, A549 tumor cells preferentially died, while the viability of normal lung fibroblasts Wi-38 remained high. Such conditions, we further designated as semi-selective for killing cancer cells. The sensitivity of cancer cells to cold plasma irradiation could vary depending on the irradiation modes used [8,44,45,46,47,48,49,50]. Previously, we demonstrated the varied sensitivity of mouse tumor cells lines to CAP with the molecular signatures of immunogenic cell death [51,52]. It was clear that the intensification of irradiation parameters could result in more significant activation of stress conditions, leading to cell death. For example, Siu et al., found that cell death of U373MG in response to helium jet CAP was a result of either apoptosis or necrosis, depending on treatment duration [53]. Indeed, in our previous work, we observed rapid cell death with a real-time monitoring technique while using severe irradiation conditions [8]. Thus, the types of cell death under soft versus severe CAP exposure might differ. Cold plasma irradiation has been studied less than routinely-used radiotherapy. In turn, for radiotherapy, it has been shown that, depending on the cell cycle phase during ionizing radiation, the radiosensitivity and mechanisms leading to cell death may differ [54,55]. While much is already known about cell death induction under severe CAP irradiation parameters, there are contradicting data for soft, semi-specific exposure. Thus far, whether autophagy is activated in response to CAP exposure has remained unclear, as has the direction of this autophagy, i.e., pro-death or pro-survival [19,56]. Considering the above, it was especially important for us to investigate the activation of autophagy in a semi-selective mode of CAP irradiation, with a focus on further clinical applications. It was assumed that parameters such as the baseline level of autophagy, drug-resistant pump activity, antioxidant level and quality of apoptosis system would be important for sensitivity to CAP.

Applying the optimal CAP regime, we analyzed the sensitivity of pairs of lung-originated cells—adenocarcinoma and normal lung fibroblasts—to direct cold atmospheric plasma irradiation. We supposed that the comparison in such pairs would be more appropriate to determining the respective special features of cancer cell and healthy cell responses to CAP treatment. The differences in the accumulation of ROS and lysosomes in CAP-treated and control tumor cells were in accordance with the data of other authors [19]. Whereas the ROS produced by direct exposure of cells to CAP are short-lived, it has been suggested that major cellular damage could be due to the secondary radicals produced. Since the plasma jet itself does not distinguish which cell it interacts with—a cancer cell or a healthy cell—we were able to assume that the dose of ROS received from CAP in healthy and tumor cells should not differ significantly. However, in the formation of secondary ROS and their elimination, the level of the cells’ antioxidant systems—which would already be different in healthy and tumor cells—would be of great importance. Indeed, in early periods after irradiation (30 min) the levels of ROS in healthy cells and tumor cells were almost the same. At the same time, we considered our finding of the activation of CatD enzymatic activity, specifically in tumor cells after CAP irradiation under semi-selective conditions, to be important, since no such activation has been shown before.

In this study, we showed that the transcriptional responses of autophagy-related genes in healthy cells after CAP irradiation were weaker than in cancer cells. However, the overall tendencies in of responses were similar, except for MAPLC3B gene expression. Activation of MAPLC3B gene expression and further processing of its protein indicated an autophagy-related process in the cell. A detailed study of the signs of autophagy in mesothelioma cells after CAP treatment was performed by Shi et al. [56]. They demonstrated the appearance of the molecular markers of autophagolysosomes LAMP1 and LC3B-II (from 2 to 8 h after irradiation), but the authors did not classify the autophagy observed as pro-death or pro-survival. Convey et al. demonstrated caspase-independent cell death of U373MG glioblastoma multiform cells after CAP exposure with lysosomes accumulation and a lack of LC3B processing [19]. In turn, Yoshikawa et al. demonstrated autophagy arising in the cells cultured with CAP-activated medium and also argued that autophagy was the death pathway of treated cells [20]. Recently, Duchesne et al. [57] showed that CAP treatment promoted macrophages’ ability to eliminate internalized bacteria by promoting the association of bacteria to LAMP-1-positive phagosomes, in which bacteria were exposed to low pH and cathepsin D hydrolase.

Autophagy can inhibit or promote cell death, depending on the internal and external environment and cell type [58]. Therefore, inhibition of autophagy in certain cases could contribute to the activation of cell death. Autophagic cell death is characterized by the massive accumulation of autophagosomes, limited chromatin condensation and lysosomal protein LC3B maturation [55]. Unique lysosomal membranes play an essential role in autophagy, and the lysosomal-associated membrane proteins LAMPs control autophagic pathways [59]. It is possible that treatment with the highly alkaline lysosomotropic compound CQ increased the number of lysosomes in response to increased cellular pH [60,61]. The potential of CQ to enhance the cytotoxic effect against tumor cells and antitumor effects in vivo has already been shown for drugs with different mechanisms of cell death [60,62,63,64]. Since the inhibition of autophagy increased the cytotoxic effect, we concluded that the basal processes of autophagy were aimed at overcoming the effects of CAP irradiation and cell survival. In addition to inhibiting autophagy, CQ could also lead to a decrease in the quality and functions of mitochondria, activating cell death machinery [41]. Indeed, we observed the suppression of the Drp1 protein, which is involved in mitochondrial fission. The fact that electron microscopy samples showed no significant differences in the number of lysosomes and autophagosomes between treated and control cells could indicate a small scale of autophagy activation during CAP irradiation. However, clarification as to whether or not the cell death induced by this combination was autophagy-dependent would need to be confirmed by more targeted approaches. For example, genetic modulation of autophagy in the cells would address this question.

## 5. Conclusions

We demonstrated the importance of optimal CAP jet operation parameters for the efficient killing of cancer cells. The CAP irradiation modes that were selective or semi-selective for the killing of tumor cells were able to activate cell death types that differed under semi-selective CAP conditions. The death of cancer cells was caspase-dependent and differed from autophagic death, even though the inhibitor of autophagy, chloroquine, increased the death of irradiated cells. We therefore proposed that a combination of CAP and chloroquine be considered as a new strategy for enhancement of the cytotoxic effects of CAP.

## Figures and Tables

**Figure 1 cells-12-00290-f001:**
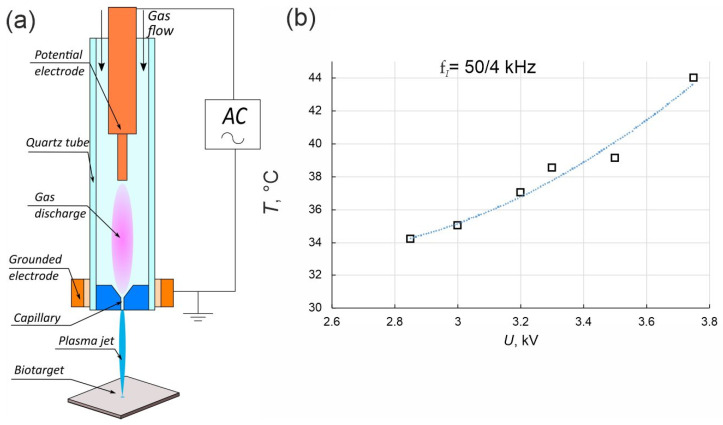
Plasma jet device and (**a**) schematic diagram of the plasma generating device. (**b**) Dependence of the temperature of the plasma jet at the point of contact with the biotarget on the applied voltage.

**Figure 2 cells-12-00290-f002:**
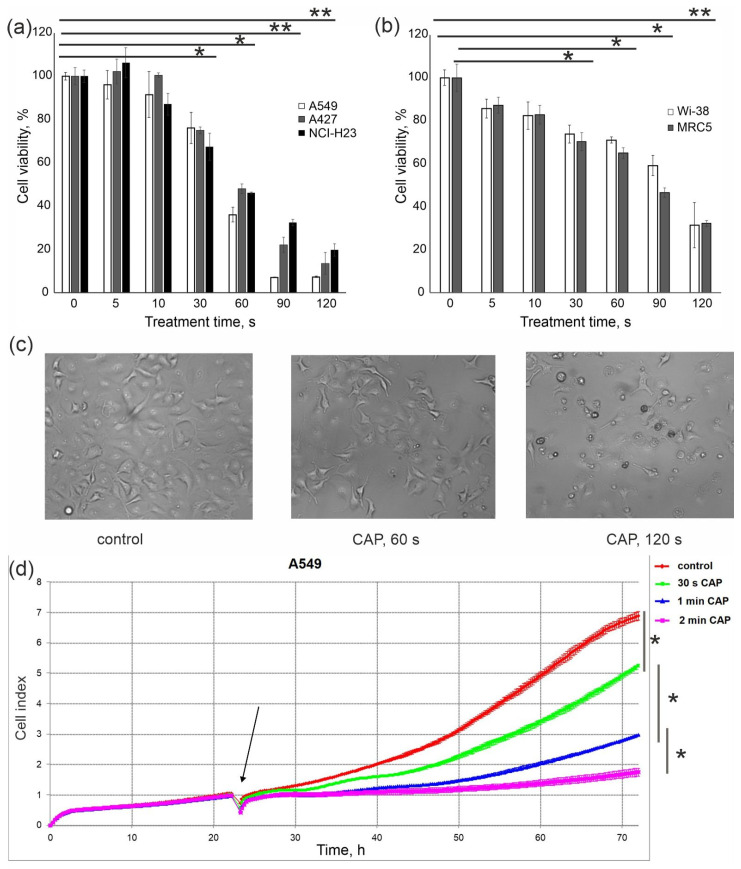
Determining cancer cell-specific irradiation conditions. Treatment conditions: voltage amplitude 3.5 kV; current frequency 50/4 kHz, working gas—helium; gas flow—9 L/min. MTT analysis data of viability for CAP-treated cancer cells (**a**) and healthy cells (**b**) performed 24 h post-irradiation. Differences are significant with * *p* < 0.05 or ** *p* < 0.05 between the control and experimental group. (**c**) Representative images of light microscopy of growing A549 cells 6 h after CAP treatment (1 min). (**d**) Real-time iCELLigence curves for A549 cells exposed to CAP. The black arrow indicates the moment of irradiation. The differences are significant with * *p* < 0.05 between the two groups.

**Figure 3 cells-12-00290-f003:**
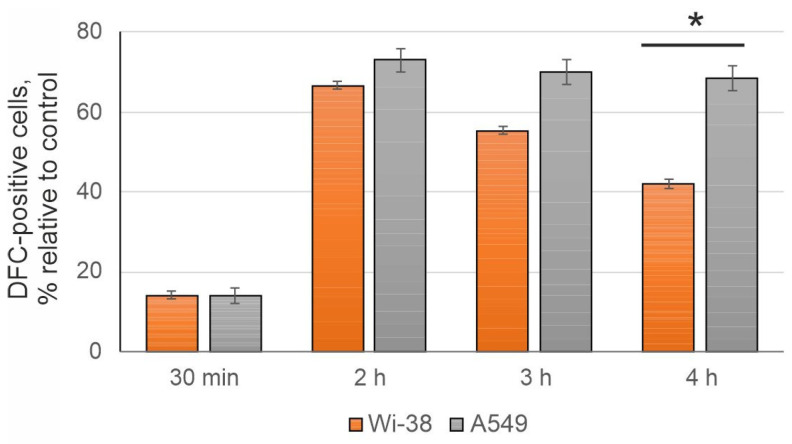
Analysis of intracellular ROS in CAP-treated cells. Flow cytometry analysis of A549 and Wi-38 cells is performed after CAP treatment (voltage amplitude 3.5 kV; current frequency 50/4 kHz, working gas—helium; gas flow—9 L/min). Cells are stained with 10 μM DCFDA (which is oxidized by ROS to DCF) for 30 min before the analysis. Data are presented as percentage of DCF-positive cells (relative to control) in FITC channel (λex = 488 nm, λem = 525 nm), % ± SD. The differences are significant with * *p* < 0.05 between two groups.

**Figure 4 cells-12-00290-f004:**
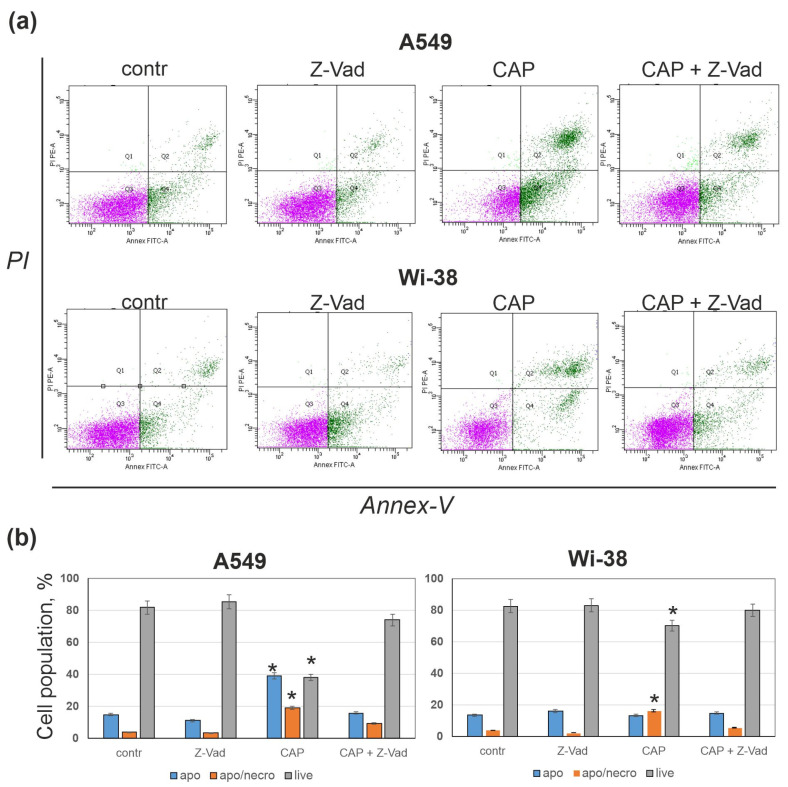
CAP induces caspase-dependent death in A549 and Wi-38 cells under semi-selective conditions of irradiation. Z-Vad (30 µM) is added to cells and, two hours later, cells are exposed to CAP for 60 s (voltage amplitude 3.5 kV; current frequency 50/4 kHz, working gas—helium; gas flow—9 L/min). 24 h later, cells are collected and stained with AnnexinV/PI. Stained cells are analyzed by flow cytometry in PE (PI, propidium iodide) and FITC (AnnexinV-FITC) channels. (**a**) A typical example of analysis. Initial gating (P1) in FSC/SSC channels was made to exclude small debris. Q1, Q2 (apo/necro), Q3 (live) and Q4 (apo) gates correspond to necrotic, late apoptotic/necrotic, live and early apoptotic cells, respectively. (**b**) The changes in the fraction of live, early apoptotic and late apoptotic/necrotic cell data are presented as average value ± SD, (*) defines the differences for *p* < 0.05 (*t*-test) with a control.

**Figure 5 cells-12-00290-f005:**
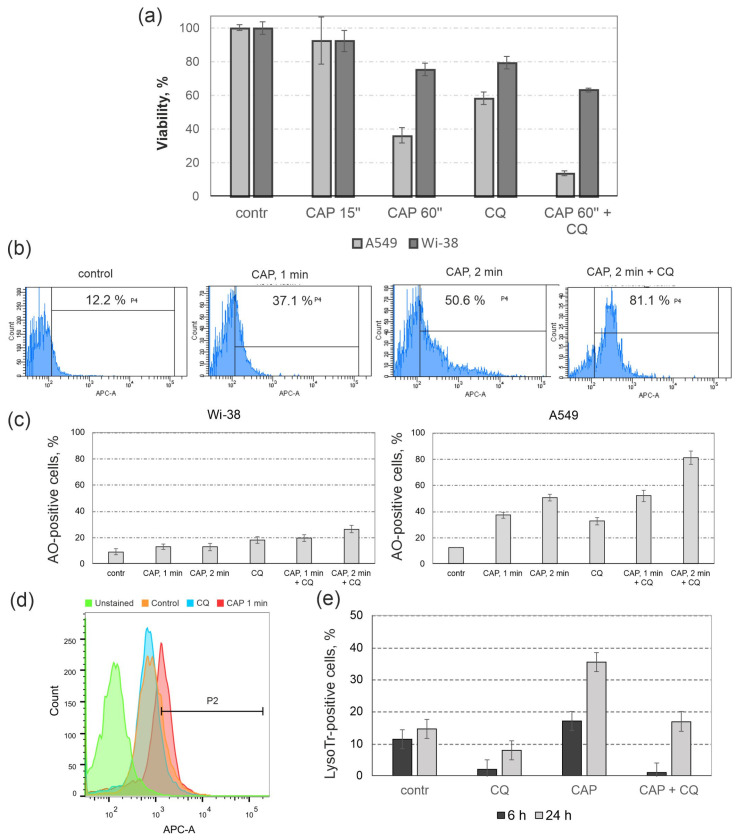
Chloroquine increases the cytotoxic effect of CAP and changes CAP-dependent lysosome biogenesis. Cells are exposed to CAP (voltage amplitude 3.5 kV; current frequency 50/4 kHz, working gas—helium; gas flow—9 L/min). Chloroquine (CQ, 20 µM) is added to the cells after CAP treatment. (**a**) Viability (MTT) assay is performed 24 h after the CAP and CQ treatment. (**b**,**c**) Analysis of lysosomes in CAP-treated cells 24 after the CAP-exposure. Cells are stained with acridine orange (AO, 1 µg/mL) and analyzed by flow cytometry in APC channel (Ex/Em = 595/660 nm). (**b**) Representative images of flow cytometry analysis of A549 cells. P4—AO-positive (lysosomes-positive) population. P4 (%) is based on the difference between the autofluorescent signal (untreated cells, no AO) and the signal from AO-treated control cells. The initial gating in FS/SS channels is made as demonstrated in Appendix A Appendix A. (**c**) Summarized flow cytometry data for A549 and Wi-38 cells. Data are presented as average value of AO-positive cells ± SD. (**d**,**e**) The analysis of A549 LysoTracker-positive cells by flow cytometry. (**d**) representative analysis 24 h post CAP exposure; P2 (%) is based on the difference between the autofluorescent signal (untreated cells, no LysoTracker) and the signal from LysoTracker-treated control cells (**e**) Summarized flow cytometry data for A549 6 h and 24 h post-CAP exposure. Lysotracker-positive cells ± SD

**Figure 6 cells-12-00290-f006:**
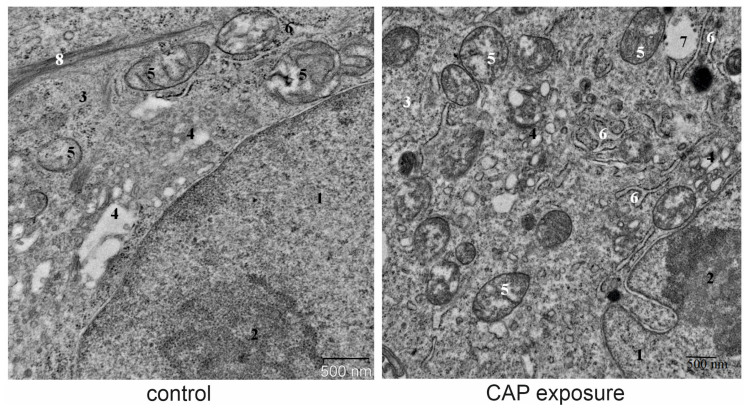
Transmission electron micrographs showing the ultrastructure of A549 cells treated CAP. Samples of control and CAP-exposed cells are prepared 24 h after CAP irradiation (60 s). Ultrathin sections. Representative example of cells fragments. 1—nucleus, 2—nucleolus, 3—cytoplasm, 4—Golgi apparatus, 5—mitochondria, 6—rough endoplasmic reticulum, 7—multivesicular body, 8—bundles of intermediate filaments.

**Figure 7 cells-12-00290-f007:**
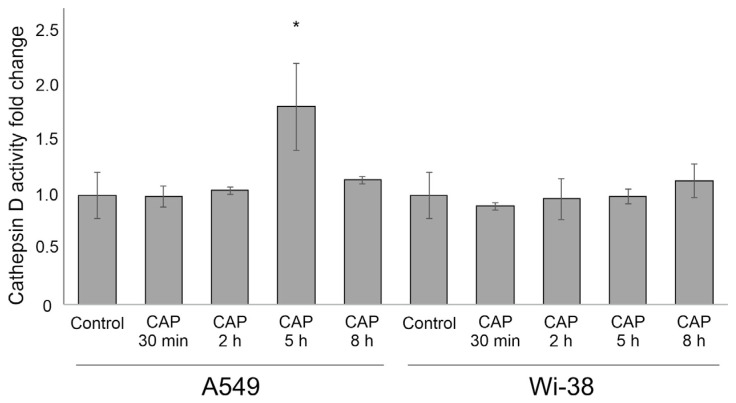
CatD activity in cancer and normal cells after CAP exposure. Analysis of cell lysates performed 30 min, 2 h, 5 h and 8 h after CAP treatment (voltage amplitude 3.5 kV; current frequency 50/4 kHz, working gas—helium; gas flow—9 L/min, 1 min treatment). RFU—relative fluorescent units. The differences are significant with * *p* < 0.05 between control and experimental groups.

**Figure 8 cells-12-00290-f008:**
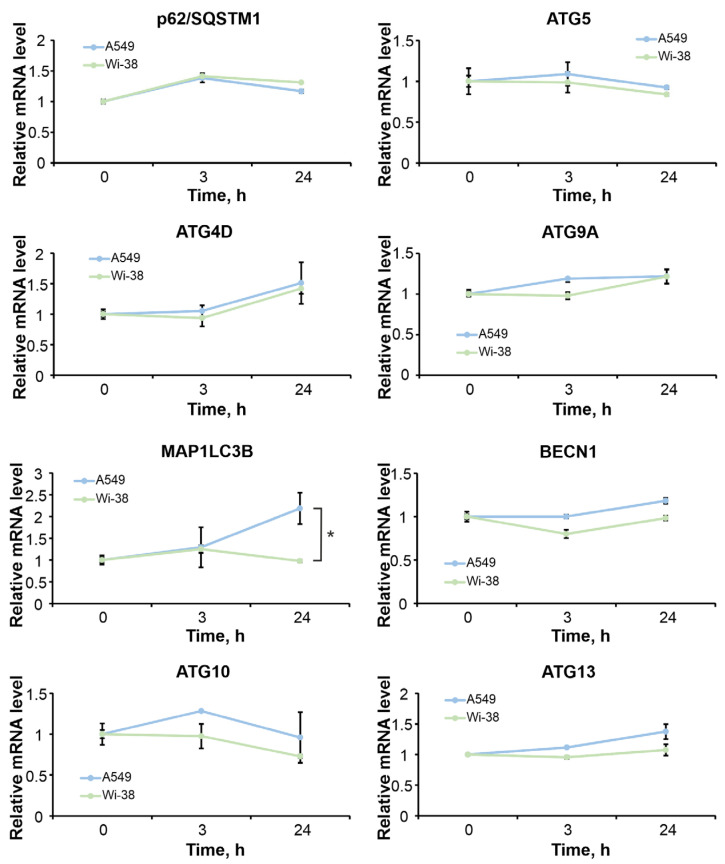
Time-dependent changes in the relative mRNA-levels of autophagy-related genes. A549 and Wi-38 cells exposed to CAP for 1 min (voltage amplitude 3.5 kV; current frequency 50/4 kGz, helium flow 9 L/min). DESeq2 normalized RNA-Seq data. Curves of individual mRNA levels in the samples are presented. The median of two independent experiments. Student’s *t*-test used for statistical analysis, (*)—*p* < 0.05 between control and experimental groups, respectively.

**Figure 9 cells-12-00290-f009:**
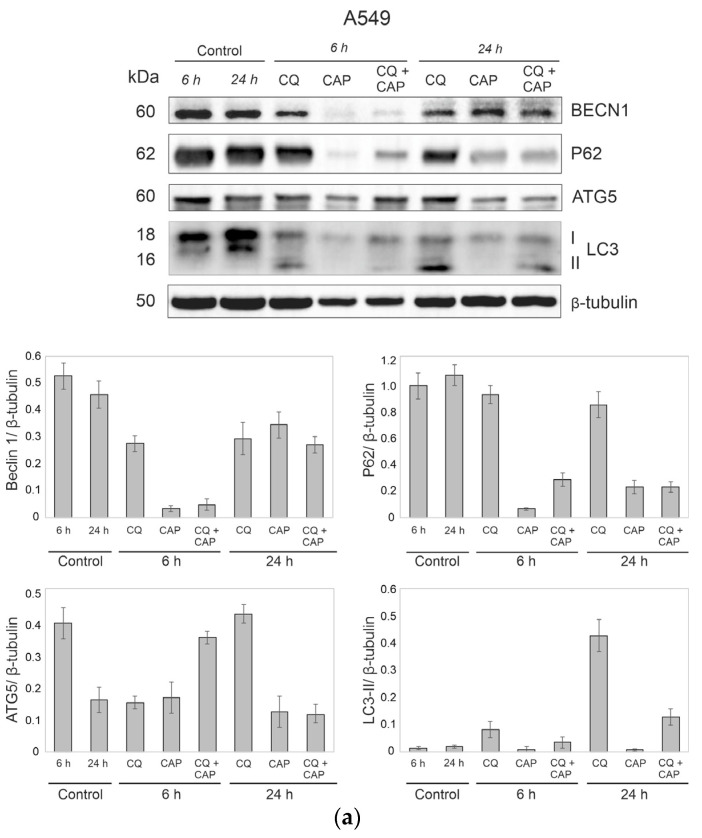
Changes in cellular proteins after CAP treatment. Representative Western blots show changes in autophagy-related proteins. Whole cell lysates are prepared for analysis using b-tubulin as a loading control. A549 (**a**) and Wi-38 (**b**) cells are exposed to CAP for 1 min (voltage amplitude 3.5 kV; current frequency 50/4 kHz, helium flow 9 L/min). CQ is added, alone or to the cells, after CAP treatment to the final concentration of 20 µM.

**Figure 10 cells-12-00290-f010:**
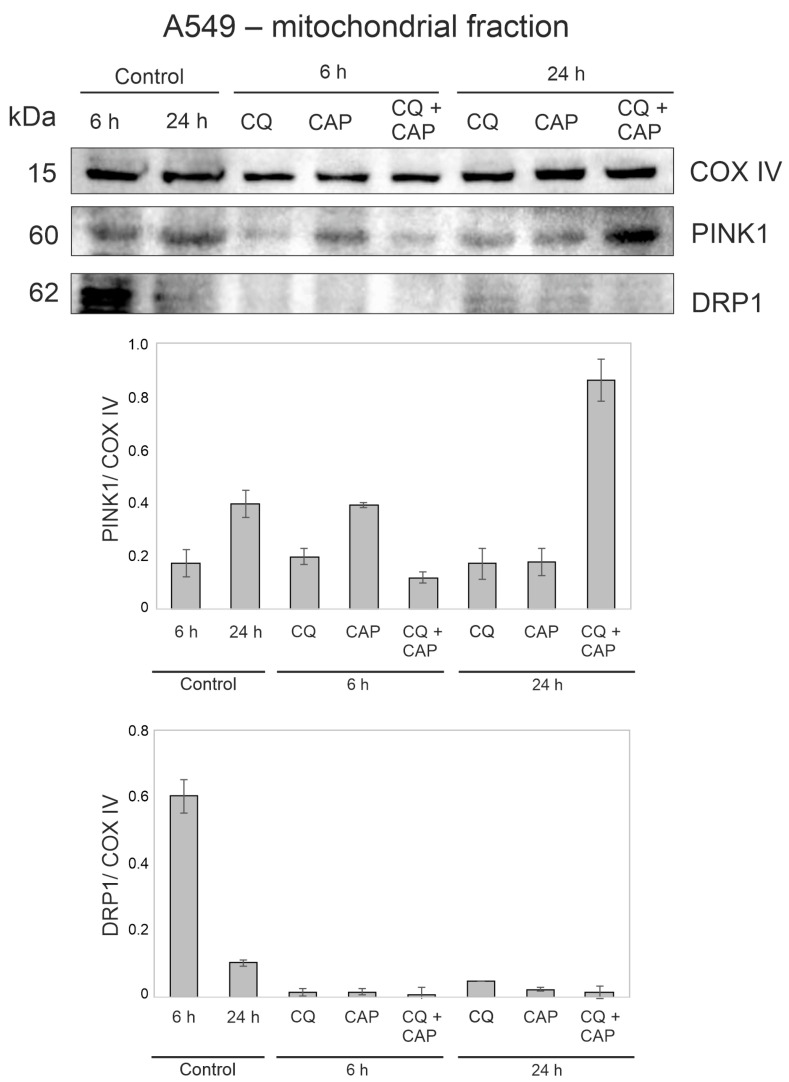
Changes in the mitochondrial proteins after CAP treatment in the A549 cells. Representative western blots showing the changes in PINK1 and Drp1 proteins. Mitochondrial fraction analyzed and COX IV used as a loading control. A549 cells exposed to CAP for 1 min (voltage amplitude 3.5 kV; current frequency 50/4 kHz, helium flow 9 L/min). CQ added alone to the cells or after CAP treatment to the final concentration, 20 µM.

## Data Availability

All data used to support the findings of this study are available from the corresponding author upon request.

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
