# Peer review of "Chloroquine Enhances Death in Lung Adenocarcinoma A549 Cells Exposed to Cold Atmospheric Plasma Jet"

_cells, 2023, doi:10.3390/cells12020290_

Round 1

Reviewer 1 Report

In this manuscript, authors used CAP to investigate the cell viability, apoptosis, intracellular ROS and autophagy. This is an interesting work. In my opinion, after a minor revision, the manuscript can be accepted.

(1) Fig. 2, list the wavelength and light density value on irradiation

(2) Fig. 4, give the percentage of living, necrotic, late and early apoptosis in the control.

(3) Beclin-1 is marker of autophagy, evaluate the effect of CPA on the expression of Beclin-1 protein.

Author Response

Reviewer 1

In this manuscript, authors used CAP to investigate the cell viability, apoptosis, intracellular ROS and autophagy. This is an interesting work. In my opinion, after a minor revision, the manuscript can be accepted.

Answer

We thank the reviewer for appreciation of our work.

Comment 1.  Fig. 2, list the wavelength and light density value on irradiation.

Answer

We thank the reviewer for this comment. We would like to emphasize that the irradiation of cell samples is carried out by a plasma jet, which is a flow of streamers (plasma formations). Therefore, in this case, it is impossible to list the wavelength and light density value on irradiation. Nevertheless, the manuscript provides all the characteristics of the plasma jet that allow to reproduce the irradiation conditions.

Only recently, studies of the effect of strong electric fields initiated during the generation of the plasma jet and their effect on cells have just begun, and the role of optical emission (individual spectral lines of OH, He, N2, H and various molecular ions, etc.) is considered minor and its role is not currently discussed.

Comment 2.  Fig. 4, give the percentage of living, necrotic, late and early apoptosis in the control.

Answer

We thank the reviewer for this comment. We modified Fig. 4 and indicated the percentage of living, late and early apoptosis in the histogram, please see new Fig. 4.

Comment 3.  Beclin-1 is marker of autophagy, evaluate the effect of CPA on the expression of Beclin-1 protein.

Answer

As recommended by the reviewer, western blot data for Beclin-1 protein has been added, please see Fig. 9.

Reviewer 2 Report

This manuscript, entitled "Chloroquine enhances autophagy-independent death in lung adenocarcinoma A549 cells exposed to cold atmospheric plasma jet", aims at investigating whether autophagy activation occurs in A549 human lung carcinoma cells and Wi-38 human lung fibroblasts after CAP exposure, and whether autophagy could contribute to CAP-induced cell death. In general, I found this article quite confusing with the feeling that the authors tried to bring together results that do not correlate all the time to each other’s. Moreover, there are quite substantial writing errors throughout the manuscript (see below, minor points).

Main points

At first, the authors identify the operative CAP settings to trigger predominantly cell death in cancer cells than in normal cells, and found that a treatment time of 60-90s was optimal (or semi-selective). Next, they used a non-selective ROS probe (H2DCFDA) to monitor ROS level in tumor and normal cells. It is unclear to me 1) why H2DCFDA was added on trypsinized cells (see lines 227-228) and why the ROS detection was assessed 3h post CAP ? Indeed, ROS level is expected to decline over time post CAP treatment and a kinetic is more informative rather than looking at one treatment time. Furthermore, and in order to assess short-live ROS, H2DCFDA can be incubated with the cells prior to CAP treatment. The main advantage of this approach is also to avoid differences in the ROS probe uptake between tumor and normal cells due to CAP-induced membrane damage. Finally, I suggest to do this experiment using also 90 s treatment time, which looks to me the most interesting treatment time because it corresponds to the biggest difference in cell viability between tumor and normal cells. And it should strengthen, or not, the conclusion made by the authors that “An elevated ROS accumulation in cancer cells after CAP irradiation positively correlated with cell death rate evaluated.” (see lines 222-223).

Next, the authors analyze if death of CAP-exposed A549 cells under semi-selective conditions is caspase-dependent. It is not clear to me why the authors performed such experiment, what is the link with the role of autophagy in the cytotoxic effects of CAP ? Indeed, while caspases are the principal mediators of the apoptotic cell death response, autophagy promotes cell survival (Tsapras and Nezis Cell Death and Differentiation (2017) 24, 1369–1379). Furthermore, why Z-Vad was added post CAP treatment and why the assay was performed at 12 h post CAP ?

While in the previous experiments, the authors observed a role of caspases in CAP-induced cell death, it is a bit strange that they decided to focus on the autophagy signatures in cells after CAP irradiation. For that, the authors used chloroquine (CQ), a well-known inhibitor of autophagy (Mauthe et al. Autophagy. 2018; 14(8): 1435–1455) in combination to acridine orange (AO). Most of the conclusions presented by the authors rely on the use of AO. My main concern is that not only AO is not specific to lysosomes but also stain nucleic acids (i.e. DNA and RNA), but moreover its concentration in the extracellular environment is important (Pierzynska-Mach et al. Cytometry Part A 85A: 729-737, 2014). In this aforementioned study, the authors used a concentration of 2.6 µM (i.e. 0.7 µg/mL), while in another study (Conway et al. Scientific Reports (2019) 9:12891), the authors used 1 µg/mL. In the present reviewed manuscript, the authors used a concentration of 1 mg/mL (1000x more concentrated compare to the two other studies). How the authors can exclude that at such high concentration, AO stains also nucleic acids ? Furthermore, the authors used flow cytometry to evaluate AO staining. But such an approach cannot discriminate within the cells which compartment is stained. I strongly suggest the authors not only to perform some fluorescence images as described in Pierzynska-Mach et al. but also to use another lysosome markers such as Lyso TrackerTM Deep Red (see Conway et al.).

In lines 284-286, it is written “Acridine orange is a fluorescent dye which is accumulated in acidic lysosomes and its red fluorescent signal reflects the amount of functionally-active lysosomes”. However, since CQ accumulates into acidic lysosomes and increase the pH due to its protonation, thereby inhibiting autophagic degradation in the lysosomes (Redman et al., Redox Biology 11 (2017) 73–81; Homewood et al., Nature Vol. 235 (1972) 50-52), how to explain the increase of red fluorescent upon combined treatment of CQ + AO (Figure 5C)? Same question regarding the effects of CAP? Are the authors convinced that they are looking at lysosomes’ activity? This is also supported by the EM micrographs showing no significant increase in amount and changes in morphology of autophagosomes and lysosomes in CAP-treated cells compared to control cells. Finally, the authors claim that “the combination of CAP irradiation and CQ significantly increases the number of lysosomes in treated cells promoting the cell death” (lines 311-312). But this is not shown. This assumption is only based on AO signal, which is highly debatable.

Next the authors focused more on specific gene expression and proteins’ level related to autophagy. Although they observed an increase in cathepsin D activity after CAP treatment in tumor cells (which is supposed to be linked to an increase in the number of lysosomes in tumor cells after CAP irradiation -lines 320-321- , but not demonstrated), the authors concluded that CAP causes minor changes in the expression of autophagy-related genes in healthy and cancer cells, and protein composition of CAP mono-treatment samples did not reveal an increase in autophagy (lines 364-365). Overall these data strongly suggest that CAP treatment does not induce an autophagy response in A549 cells. But it is not clear to me how to link the increase of lysosomes induced by CAP in A549 (claim by the authors in lines 294-295) and the fact that autophagy is not or poorly increase in CAP-treated A549 cells?

Finally, the cytotoxicity of CQ in CAP-treated A549 cells can be essentially due to a decrease in mitochondrial quality and bioenergetic function in A459 cells. Indeed, it has been shown that inhibition of autophagy with chloroquine decreases mitochondrial quality and bioenergetic function in primary neurons (Redmann et al., Redox Biology 11 (2017) 73–81). This should be investigated by the authors.

 Minor points

                 Line 94: The ATCC # for Wi-38 is not CCL-185 but CCL-75

                Line 102: “A cells” is not correct

                Line 109: What is the cell viability at 72h? It is generally informative to have different time points to assess cell viability by MTT assay.

                Line 132: softwares

Line 146: Is it really 2x105 cells per well in 96-well plates?? It seems an extremely high number of cells for such as a very small surface.

Line 158: acridine orange (2.7 mM) but in line 274 [AO]=4.05 mM ???

Line 160: FITC fluorescent channel but in line 275  “analyzed by flow cytometry in APC channel (Ex/Em = 595/660 nm)”??

Line 164: It is strange to use L-15 medium at this step. What is the explanation?

Line 165: I doubt that the cells were precipitated but they were collected by centrifugation and cell pellets resuspended in 4% PAF.

Line 246: For sure, A549 cells were not 60 min exposed to CAP

                Line 325: Indicate the treatment time of CAP.

                Line 334: Why the authors have chosen 3h while in Figure 7 they performed the experiments at 2h and 5h post CAP??

                Line 361: It is indicated 3h and 24h while in Figure 9, the indicated times are 6h and 24h??  

                Line 419: I guess it is “lack” and not “luck”

Author Response

Reviewer 2

This manuscript, entitled "Chloroquine enhances autophagy-independent death in lung adenocarcinoma A549 cells exposed to cold atmospheric plasma jet", aims at investigating whether autophagy activation occurs in A549 human lung carcinoma cells and Wi-38 human lung fibroblasts after CAP exposure, and whether autophagy could contribute to CAP-induced cell death. In general, I found this article quite confusing with the feeling that the authors tried to bring together results that do not correlate all the time to each other’s. Moreover, there are quite substantial writing errors throughout the manuscript (see below, minor points).

Answer

We thank the reviewer for a very detailed analysis of our manuscript. We are very grateful for the opportunity to correct the points that were raised in the course of the revision. We have tried to make our hypotheses clearer in the revised version of the manuscript. Moreover, we have added a significant part of experimental data to support our hypothesis. We have also tried to correct writing errors throughout the manuscript. A native English speaker has edited the text.

Main points

Comment

At first, the authors identify the operative CAP settings to trigger predominantly cell death in cancer cells than in normal cells, and found that a treatment time of 60-90s was optimal (or semi-selective). Next, they used a non-selective ROS probe (H2DCFDA) to monitor ROS level in tumor and normal cells. It is unclear to me 1) why H2DCFDA was added on trypsinized cells (see lines 227-228) and why the ROS detection was assessed 3h post CAP ? Indeed, ROS level is expected to decline over time post CAP treatment and a kinetic is more informative rather than looking at one treatment time. Furthermore, and in order to assess short-live ROS, H2DCFDA can be incubated with the cells prior to CAP treatment. The main advantage of this approach is also to avoid differences in the ROS probe uptake between tumor and normal cells due to CAP-induced membrane damage. Finally, I suggest to do this experiment using also 90 s treatment time, which looks to me the most interesting treatment time because it corresponds to the biggest difference in cell viability between tumor and normal cells. And it should strengthen, or not, the conclusion made by the authors that “An elevated ROS accumulation in cancer cells after CAP irradiation positively correlated with cell death rate evaluated.” (see lines 222-223).

Answer

During the preliminary stage of our work, we have tested several techniques of H2DCFDA cell staining for subsequent analysis by flow cytometry. The most reproducible results for long-term (more than 1 h) experimental points were obtained using the technique described in: Bakewell S, Conde I, Fallah Y, McCoy M, Jin L, Shajahan-Haq AN. Inhibition of DNA Repair Pathways and Induction of ROS Are Potential Mechanisms of Action of the Small Molecule Inhibitor BOLD-100 in Breast Cancer. Cancers (Basel). 2020 Sep 16;12(9):2647. doi: 10.3390/cancers12092647. PMID: 32947941; PMCID: PMC756376.

Considering the reviewer's comments, we modified the protocol of analysis. Since it was important to use cells treated uniformly to compare results, we added H2DCFDA to the cells immediately after irradiation for the 30 min sample and 30 min before flow cytometry for the remaining samples (Please, see methods, L. 143-144).

Considering the reviewer's comments, we have added earlier (30 min and 2 h) and later (4 h) time points for analysis (please, see new Fig. 3). Moreover, a discussion of the results obtained has been added to the "discussion" section (please, see L 513-531): Whereas the ROS produced by direct exposure of cells to CAP are short-lived, it is suggested that the major cellular damage is due to the secondary radicals produced. Since the plasma jet itself does not distinguish which cell it interacts with - a cancer cell or a healthy cell, we can assume that the dose of ROS received from CAP in healthy and tumor cells should not differ significantly. But in the formation of secondary ROS and their elimination, the level of the cell's antioxidant system, which will already be different in healthy and tumor cells, is of great importance. Indeed, in early periods after irradiation (30 min) the level of ROS in healthy and tumor cells was almost the same.

Comment

Next, the authors analyze if death of CAP-exposed A549 cells under semi-selective conditions is caspase-dependent. It is not clear to me why the authors performed such experiment, what is the link with the role of autophagy in the cytotoxic effects of CAP? Indeed, while caspases are the principal mediators of the apoptotic cell death response, autophagy promotes cell survival (Tsapras and Nezis Cell Death and Differentiation (2017) 24, 1369–1379). Furthermore, why Z-Vad was added post CAP treatment and why the assay was performed at 12 h post CAP ?  

Answer

As recommended by the reviewer, we performed experiments where Z-vad was added 2 h before CAP exposure (see new Fig. 4). The time point for the analysis was 24 h after the CAP exposure.

We have added the following justification to the main text (see L. 313-319): “There is growing evidence that in specific contexts autophagy can indeed facilitate cell death. Cell death by autophagy can be defined as cell death that is independent of apoptosis, which can be blocked by pharmacological or genetic inhibition of autophagy and results in an increase in autophagic flux (completion of autophagy), rather than only increased detection of markers of autophagy [18]. In the case of therapeutic drugs or treatments, pro-survival autophagy can be considered a negative factor and pro-death autophagy a positive factor”.

Comment

While in the previous experiments, the authors observed a role of caspases in CAP-induced cell death, it is a bit strange that they decided to focus on the autophagy signatures in cells after CAP irradiation.

For that, the authors used chloroquine (CQ), a well-known inhibitor of autophagy (Mauthe et al. Autophagy. 2018; 14(8): 1435–1455) in combination to acridine orange (AO). Most of the conclusions presented by the authors rely on the use of AO. My main concern is that not only AO is not specific to lysosomes but also stain nucleic acids (i.e. DNA and RNA), but moreover its concentration in the extracellular environment is important (Pierzynska-Mach et al. Cytometry Part A 85A: 729-737, 2014). In this aforementioned study, the authors used a concentration of 2.6 µM (i.e. 0.7 µg/mL), while in another study (Conway et al. Scientific Reports (2019) 9:12891), the authors used 1 µg/mL. In the present reviewed manuscript, the authors used a concentration of 1 mg/mL (1000x more concentrated compare to the two other studies). How the authors can exclude that at such high concentration, AO stains also nucleic acids ? Furthermore, the authors used flow cytometry to evaluate AO staining. But such an approach cannot discriminate within the cells which compartment is stained. I strongly suggest the authors not only to perform some fluorescence images as described in Pierzynska-Mach et al. but also to use another lysosome markers such as Lyso TrackerTM Deep Red (see Conway et al.).

Answer

The following reasoning has been added to the main text (see L 310-321): “Recently, it has been demonstrated that ROS can induce autophagy leading to the death of cancer cells, meaning that autophagy may contribute to CAP-induced cell death [20]. Therefore, we hypothesized that CAP-dependent ROS accumulation may also stimulate autophagic cell death in treated cells. There is growing evidence that in specific contexts autophagy can indeed facilitate cell death. Cell death by autophagy can be defined as cell death that is independent of apoptosis, which can be blocked by pharmacological or genetic inhibition of autophagy and results in an increase in autophagic flux (completion of autophagy), rather than only increased detection of markers of autophagy [18]. In the case of therapeutic drugs or treatments, pro-survival autophagy can be considered a negative factor and pro-death autophagy a positive factor. When developing an antitumor approach, the use of the most appropriate partner drug can greatly enhance the therapeutic effect. Such a partner could be a drug that inhibits pro-survival autophagy”.

We strongly agree with the reviewer's comments concerning acridine orange. The experiments with LysoTrecker Red have been added (please, see Fig. 5 d, e). LysoTracker Red DND-99 is a cell-permeable red fluorescent dye that stains acidic compartments within a cell, such as lysosomes. Moreover, the statement “Thus, the addition of CQ to the CAP-irradiated cells results in the substantial increase of lysosomes-related signals compared to mono-treatment in the both cancer and normal cells” was removed from the text.

In our study, we used a AO concentration of 1 µg/mL; there was a typo in the original version. Kindly accept our apologies.

The modified conclusion of this part of the study is as follows: “Thus, the combination of CAP irradiation and CQ significantly promotes cell death, despite the fact that CAP weakly stimulates the increase of lysosomes” (L. 385-386).

Comment

In lines 284-286, it is written “Acridine orange is a fluorescent dye which is accumulated in acidic lysosomes and its red fluorescent signal reflects the amount of functionally-active lysosomes”. However, since CQ accumulates into acidic lysosomes and increase the pH due to its protonation, thereby inhibiting autophagic degradation in the lysosomes (Redman et al., Redox Biology 11 (2017) 73–81; Homewood et al., Nature Vol. 235 (1972) 50-52), how to explain the increase of red fluorescent upon combined treatment of CQ + AO (Figure 5C)? Same question regarding the effects of CAP? Are the authors convinced that they are looking at lysosomes’ activity? This is also supported by the EM micrographs showing no significant increase in amount and changes in morphology of autophagosomes and lysosomes in CAP-treated cells compared to control cells. Finally, the authors claim that “the combination of CAP irradiation and CQ significantly increases the number of lysosomes in treated cells promoting the cell death” (lines 311-312). But this is not shown. This assumption is only based on AO signal, which is highly debatable.

Answer

We thank the reviewer for this valuable comment. Since LysoTracker (LT) probes are highly selective for acidic organelles, the adding of CQ to the samples shifts the lysosomal рН from 4.5 to 7.4 and we detect this as decreasing the LT-positive cells population (to control). However, we observed the increase of LT-positive cells in the CAP-treated samples (please, see new Fig.5).

Comment. 

Next the authors focused more on specific gene expression and proteins’ level related to autophagy. Although they observed an increase in cathepsin D activity after CAP treatment in tumor cells (which is supposed to be linked to an increase in the number of lysosomes in tumor cells after CAP irradiation -lines 320-321- , but not demonstrated), the authors concluded that CAP causes minor changes in the expression of autophagy-related genes in healthy and cancer cells, and protein composition of CAP mono-treatment samples did not reveal an increase in autophagy (lines 364-365). Overall these data strongly suggest that CAP treatment does not induce an autophagy response in A549 cells. But it is not clear to me how to link the increase of lysosomes induced by CAP in A549 (claim by the authors in lines 294-295) and the fact that autophagy is not or poorly increase in CAP-treated A549 cells?

Answer

Indeed, we agree with the reviewer that it is incorrect to claim that "Overall these data strongly suggest that CAP treatment does not induce an autophagy response in A549 cells."  We just like to emphasize the absence of an autophagy-related death response.

The statement has been revised as follows: “Protein composition of CAP mono-treatment samples revealed slightly increase in autophagy,” (see L.437-438). We hope, that new data helps to support our conclusions.

Comment. 

Finally, the cytotoxicity of CQ in CAP-treated A549 cells can be essentially due to a decrease in mitochondrial quality and bioenergetic function in A459 cells. Indeed, it has been shown that inhibition of autophagy with chloroquine decreases mitochondrial quality and bioenergetic function in primary neurons (Redmann et al., Redox Biology 11 (2017) 73–81). This should be investigated by the authors.

Answer

We analyzed changes in mitochondrial quality control proteins (Please, see Fig. 10,  and L. 461-480).

 Minor points

          Line 94: The ATCC # for Wi-38 is not CCL-185 but CCL-75

Answer

We apologize for the mistakes that were made by us in the initial submission and cordially thank the reviewer for pointing them out.

            Line 102: “A cells” is not correct

This was corrected.

            Line 109: What is the cell viability at 72h? It is generally informative to have different time points to assess cell viability by MTT assay.

Answer

We've added real-time viability curves (please, see Fig. 2 d)

            Line 132: softwares

This was corrected.

Line 146: Is it really 2x105 cells per well in 96-well plates?? It seems an extremely high number of cells for such as a very small surface.

Answer

That's a typo. Correct is 2x103 (please, see L. 183).

Line 158: acridine orange (2.7 mM) but in line 274 [AO]=4.05 mM ???

Answer

Corrected. Stock solution of AO (2.7 mM) was used to stain cells at final concentration 1 µg/mL (see L. 195).

Line 160: FITC fluorescent channel but in line 275 “analyzed by flow cytometry in APC channel (Ex/Em = 595/660 nm)”??

Answer

This was corrected in Methods “APC fluorescent channel”. (see L.197)

Line 164: It is strange to use L-15 medium at this step. What is the explanation?

Answer

We used the L-15 medium at this stage in order to damage the cells little as possible. The serum in this formulation is destined to inactivate trypsin. When preparing cell and tissue samples for TEM, we always use culture media or Hanks' solution as buffer solutions, since the cells are “comfortable” with their composition, and it well preserves the cell structure until complete fixation.

Culture media and Hank's solution also have greater buffering capacity than standard buffers, which also contributes to better fixation.

Line 165: I doubt that the cells were precipitated but they were collected by centrifugation and cell pellets resuspended in 4% PAF.

Answer

This was corrected, , please see L 209-210.

Line 246: For sure, A549 cells were not 60 min exposed to CAP

Answer

60 seconds. This was corrected, please see L. 297.

            Line 325: Indicate the treatment time of CAP.

This was corrected, please see L. 400.

            Line 334: Why the authors have chosen 3h while in Figure 7 they performed the experiments at 2h and 5h post CAP??

Answer

Indeed, in some experiments, time points of 2h or 3h and 5h or 6 h were chosen. In some experiments, this was due to technical aspects of sample preparation.

            Line 361: It is indicated 3h and 24h while in Figure 9, the indicated times are 6h and 24h?? 

This was corrected (6h), see L. 435.

            Line 419: I guess it is “lack” and not “luck”

Answer

This was corrected.

Reviewer 3 Report

Summary:

In this manuscript, the authors investigated the effect of cold atmospheric plasma jet -/+ chloroquine on A549 lung cancer cell line and Wi-38 fibroblast cell line. Overall, the present findings are interesting, yet there are some issues needed to be addressed before accepting this manuscript.

Major points:

1. Only one lung cancer and one fibroblast cell lines were used. Nonetheless, the authors concluded that the results stemming from one cell line/each could be generalized to lung cancer cells and normal cells. At least, the cell viability assay should be confirmed in several lung cancer and normal cell lines.

2. Figure 2 demonstrated that 60 sec post-CAP dramatically affected the cell viability in A549 but not Wi-38 cells. In contrast, Figure.3 did not show corresponding difference in ROS levels as noted by the authors. However, they concluded that “an elevated ROS accumulation in cancer cells after CAP irradiation positively correlated with cell death rate”. This should be corrected to match the actual results which were different after 60 and 90 sec.

3. In line 279, the authors concluded that “The finding that inhibition of autophagy enhances CAP-initiated cell death, indirectly indicates the involvement of autophagy in CAP-dependent cytotoxicity”. Nonetheless, this conclusion is not precise because it has been previously shown that chloroquine acts through several autophagy-independent mechanisms. Notably, the consensus in the autophagy research field (as outlined by Klionsky et al Autophagy, 2021) that pharmacological non-selective inhibitor as chloroquine can not be used  alone to infer conclusions about the role/involvement of autophagy. The authors need to carefully interpret their results.

4. Unlike the cell viability/ROS assays carried out after 1 and 2 min post-CAP. Cathepsin D activity was assessed 2 and 5 h after the CAP treatment. What is the status of Cathepsin D activity in the earlier time points? Why didn’t the authors stick to the original time points to match the molecular results with the phenotypic/viability ones?

5. As in point 4, the authors performed RNA-Seq on A459 332 and Wi-38 cells irradiated by CAP under semi-selective conditions (1 min), and then after 3 and 24 hours. Additionally, the definition of the differentially expressed genes is not stated? What are the DEGs genes? Did the authors perform gene set enrichment analysis? Are the RNA-Seq dataset deposited on GEO database? If yes, what is its accession number?

6. Figure.9, there is a single control for 6 h and 24h post-CAP administration. Cell lysates from the controls of 6h and controls of the 24h should be both provided.

7. The authors should also explain the reason for showing different readouts with different assays without providing an explanation for the reason. This applies for almost all assays with some carried out at 1,2 min (cell viability, ROS, AO), 3 and 24h (RNA-Seq) and 2 and 24h (western blot).

8.  In Figure.9, the protein loading of A549 lysates is not even and it is not possible to interpret the results with such uneven protein loading.

9. The title of the manuscript is not precise. “Chloroquine enhances autophagy-independent death in lung adenocarcinoma A549 cells exposed to cold atmospheric plasma jet”. Indeed, the authors interpret the troublesome figure.9 that chloroquine-CAP combination modulated autophagy markers. Nonetheless, they did not test whether the cell death induced by this combination is autophagy dependent or not. Genetic modulation of autophagy would address this question. This should be discussed/clarified within the manuscript.

Minor points:

1.  Before describing the results of the effect of CAP on apoptosis, the authors prepared the reader to visualize the effect on autophagy. Please see the paragraph below and position it in the proper place. “Recently, it has been demonstrated that ROS can induce autophagy leading to the 231 death of cancer cells, meaning that autophagy may contribute to CAP-induced cell death 232 [19]. Therefore, we hypothesized that CAP-dependent ROS accumulation may stimulate 233 autophagic cell death in treated cells.”

2.  Few grammar and spelling mistakes (e.g. synergism) have been noticed throughout the manuscript and should be corrected.

Author Response

Reviewer 3

In this manuscript, the authors investigated the effect of cold atmospheric plasma jet -/+ chloroquine on A549 lung cancer cell line and Wi-38 fibroblast cell line. Overall, the present findings are interesting, yet there are some issues needed to be addressed before accepting this manuscript.

Major points:

  1. Only one lung cancer and one fibroblast cell lines were used. Nonetheless, the authors concluded that the results stemming from one cell line/each could be generalized to lung cancer cells and normal cells. At least, the cell viability assay should be confirmed in several lung cancer and normal cell lines.

Answer

We thank the reviewer for this сomment. Two lung tumor lines and one line of healthy lung cells were added to the study (please, see new Fig. 1).

  1. Figure 2 demonstrated that 60 sec post-CAP dramatically affected the cell viability in A549 but not Wi-38 cells. In contrast, Figure.3 did not show corresponding difference in ROS levels as noted by the authors. However, they concluded that “an elevated ROS accumulation in cancer cells after CAP irradiation positively correlated with cell death rate”. This should be corrected to match the actual results which were different after 60 and 90 sec.

Answer

This experiment was corrected according to Reviewer’s 2 recommendation (Please, see Fig. 3 new). We believe that now it is more clear that ROS are accumulated in cancer cells.

  1. In line 279, the authors concluded that “The finding that inhibition of autophagy enhances CAP-initiated cell death, indirectly indicates the involvement of autophagy in CAP-dependent cytotoxicity”. Nonetheless, this conclusion is not precise because it has been previously shown that chloroquine acts through several autophagy-independent mechanisms. Notably, the consensus in the autophagy research field (as outlined by Klionsky et al Autophagy, 2021) that pharmacological non-selective inhibitor as chloroquine can not be used alone to infer conclusions about the role/involvement of autophagy. The authors need to carefully interpret their results.

Answer

The statement “The finding that inhibition of autophagy enhances CAP-initiated cell death, indirectly indicates the involvement of autophagy in CAP-dependent cytotoxicity” has been corrected (Please, see L. 345-346): The finding that inhibition of autophagy enhances CAP-initiated cell death, may indirectly indicate the involvement of autophagy in CAP-dependent cytotoxicity. We have also added experiments on mitochondrial quality control proteins, which can also be modulated by CQ (See Fig. 11).

  1. Unlike the cell viability/ROS assays carried out after 1 and 2 min post-CAP. Cathepsin D activity was assessed 2 and 5 h after the CAP treatment. What is the status of Cathepsin D activity in the earlier time points? Why didn’t the authors stick to the original time points to match the molecular results with the phenotypic/viability ones?

Answer

We believe there was a misunderstanding here. The irradiation time in all experiments lies in the range of 1-2 minutes, and the analysis is performed not earlier than 30 min after irradiation. Nevertheless, we added a 30 min point to the CatD activity study (please, see new Fig. 7).

  1. As in point 4, the authors performed RNA-Seq on A459 332 and Wi-38 cells irradiated by CAP under semi-selective conditions (1 min), and then after 3 and 24 hours. Additionally, the definition of the differentially expressed genes is not stated? What are the DEGs genes? Did the authors perform gene set enrichment analysis? Are the RNA-Seq dataset deposited on GEO database? If yes, what is its accession number?

Answer

Thank you for pointing this out. Indeed, this paper presents selective results of a full-transcriptome study of cells subjected irradiation by CAP. In chapter 3.6. (L. 146-159) and in Figure 8 legend. We are representing some gene expression results -- Relative mRNA level changes with time of incubation after irradiation Figure 8. It should be noted here that the presentation of relative mRNA levels does not require a differential analysis of gene expression (DEGs -- how the Reviewer shortened it). Moreover, the graphs in Figure 8 show confidence intervals for individual points, which can be used to easily assess the significance of differences in mRNA levels for each time-point under study.

With regard to depositing full transcriptome analysis data in public archives (SRA/GEO), the following should be said. In this work, the data of the NGS-analysis are used as confirming, secondary results, which do not significantly affect the conclusions of the work. That is actually presented in the conclusion to the section 3.6: "However, no significant changes in the other autophagy-related genes have been found to unequivocally state the activation of autophagy. Thus, we can conclude that CAP causes minor changes in the expression of autophagy-related genes in healthy and cancer cells."

Therefore, there is no need to deposit the entire array of full transcriptomic NGS data in the SRA archive. With that, we are currently planning a separate publication based mainly on changes in the cell transcriptome under irradiation. Therefore, we consider the deposition of NGS-data for this article premature.

  1. 9, there is a single control for 6 h and 24h post-CAP administration. Cell lysates from the controls of 6h and controls of the 24h should be both provided.

Answer

In the new version Fig. 9 provides the controls of 6h and controls of the 24h (for A549 and Wi-38).

  1. The authors should also explain the reason for showing different readouts with different assays without providing an explanation for the reason. This applies for almost all assays with some carried out at 1,2 min (cell viability, ROS, AO), 3 and 24h (RNA-Seq) and 2 and 24h (western blot).

Answer

There is a misunderstanding in the question similar to question 4. The irradiation time in all experiments lies in the range of 1-2 minutes, and the analysis is performed not earlier than 30 min after irradiation. Indeed, time points 3 and 24h were used for RNA-Seq and 6 and 24h for western blot. We assumed that it takes time for the protein content to change when the amount of RNA changes.

  1. In Figure.9, the protein loading of A549 lysates is not even and it is not possible to interpret the results with such uneven protein loading.

Answer

Additional experiments have been done to address the reviewer's comment. Please, see modified Fig. 9. As it was recommended by another reviewer, the Beclin-1 analysis has also been added (Please, see new Fig. 9).

  1. The title of the manuscript is not precise. “Chloroquine enhances death in lung adenocarcinoma A549 cells exposed to cold atmospheric plasma jet”. Indeed, the authors interpret the troublesome figure.9 that chloroquine-CAP combination modulated autophagy markers. Nonetheless, they did not test whether the cell death induced by this combination is autophagy dependent or not. Genetic modulation of autophagy would address this question. This should be discussed/clarified within the manuscript.

Answer

The title was modified: Chloroquine enhances death in lung adenocarcinoma A549 cells exposed to cold atmospheric plasma jet”.

According to reviewer’s comment we added to Discussion: “However, to clarify whether the cell death induced by this combination is autophagy dependent or not need to be confirmed by more targeted approaches. For example, the genetic modulation of autophagy in the cells would address this question” (see L. 558-561).

Minor points:

  1. Before describing the results of the effect of CAP on apoptosis, the authors prepared the reader to visualize the effect on autophagy. Please see the paragraph below and position it in the proper place. “Recently, it has been demonstrated that ROS can induce autophagy leading to the 231 death of cancer cells, meaning that autophagy may contribute to CAP-induced cell death 232 [19]. Therefore, we hypothesized that CAP-dependent ROS accumulation may stimulate 233 autophagic cell death in treated cells.”

Answer

This statement has been moved throughout the text (please, see L. 310-313).

  1. Few grammar and spelling mistakes (e.g. synergism) have been noticed throughout the manuscript and should be corrected.

Answer

We have tried to correct grammatical and stylistic errors. A native English speaker has edited the text.

Round 2

Reviewer 2 Report

Dear authors

I appreciate the extensive amount of work that was added to this revised version, and the fact that you fully fulfilled my requests. I think it adds a better value to your manuscript. I just have a slight problem with Figure 5. Indeed, how was set P4 in Figure 5B? Is it based on the difference between the autofluorescent signal (untreated cells, no AO) and the signal from AO-treated cells? If so, it should be mentioned in the figure caption. Furthermore, it will be worth setting all Y-axes to the same scale (related to Ctr). Same question for Figure 5d. How were the settings defined to calculate a % of positive cells? Finally, why the % of AO-positive cells increased after CQ treatment while conversely it decreases using LysoTracker Red, while both compounds stain for acidic compartments? I didn't get the explanation. Could you clarify these points?

Author Response

Point to point response to Reviewer2.

I appreciate the extensive amount of work that was added to this revised version, and the fact that you fully fulfilled my requests. I think it adds a better value to your manuscript.

Answer

We thank the reviewer for appreciation of our work.

Comment

I just have a slight problem with Figure 5. Indeed, how was set P4 in Figure 5B? Is it based on the difference between the autofluorescent signal (untreated cells, no AO) and the signal from AO-treated cells? If so, it should be mentioned in the figure caption. Furthermore, it will be worth setting all Y-axes to the same scale (related to Ctr). Same question for Figure 5d. How were the settings defined to calculate a % of positive cells?

Answer

To explain the choose of P4 population we have added figure S2 in Supplementary. We have also described P4 in the legend of Fig. 5.  Indeed, it based on the difference between the autofluorescent signal (untreated cells, no AO) and the signal from AO-treated cells. The same way for 5d (please, see S2 and figure 5 caption). Figure 5d has been modified as requested by the reviewer.

Converting Figure 5b to Figure 5d style (Y-axes to the same scale, FlowJo software) worsens the perception of the results (superimposing 5 colors simultaneously). Therefore, we have added figure S2 to the supplementary.

Comment

 Finally, why the % of AO-positive cells increased after CQ treatment while conversely it decreases using LysoTracker Red, while both compounds stain for acidic compartments? I didn't get the explanation. Could you clarify these points?

Answer

We suppose that such differences are due to the different sensitivity of AO and LysoTr to the acidic environment. AO possible to stain light-acidic compartments: a number of papers have shown, using AO staining, that CQ increases the formation of Acidic vesicular organelles (AVOs) (Sasaki, K., Tsuno, N.H., Sunami, E. et al. , 2010, https://doi.org/10.1186/1471-2407-10-370) and the treatment with 50 μM CQ for 24 h induces the increase of the acidic compartments in M14 melanoma cells (Agostinelli E, et al., 2014, https://doi.org/10.3892/ijo.2014.2502). Therefore, your question probably deserves a special study.  In any case, it was important for our study to show an increase in acidic organelles after CAP treatment. Both experiments - with AO and lysotracker showed an increase in acidic vesicles after CAP treatment.